# Parallel mechanisms signal a hierarchy of sequence structure violations in the auditory cortex

Sara Jamali[1], Sophie Bagur[1], Enora Bremont[1], Timo Van Kerkoerle[2,3], Stanislas Dehaene[2,3], Brice Bathellier[1]*

[1]Université Paris Cité, Institut Pasteur, AP-HP, Inserm, Fondation Pour l'Audition, Institut de l'Audition, IHU reConnect, Paris, France; [2]Université Paris Saclay, INSERM, CEA, Cognitive Neuroimaging Unit, NeuroSpin Center, Paris, France; [3]Collège de France, PSL University, Paris, France

## eLife Assessment

This **important** study advances our understanding of the way neurons in the auditory cortex of mice respond to unpredictable sounds. Through the use of state-of-the-art recording methods, **compelling** evidence is provided that responses to local and global violations in sound sequences are prediction errors and not simply the consequence of stimulus-specific adaptation. Although the cell-type-specific results are intriguing, further work is needed to establish their reliability.

*For correspondence:
brice.bathellier@cnrs.fr

**Abstract** The brain predicts regularities in sensory inputs at multiple complexity levels, with neuronal mechanisms that remain elusive. Here, we monitored auditory cortex activity during the local-global paradigm, a protocol nesting different regularity levels in sound sequences. We observed that mice encode local predictions based on stimulus occurrence and stimulus transition probabilities, because auditory responses are boosted upon prediction violation. This boosting was due to both short-term adaptation and an adaptation-independent surprise mechanism resisting anesthesia. In parallel, and only in wakefulness, VIP interneurons responded to the omission of the locally expected sound repeat at the sequence ending, thus providing a chunking signal potentially useful for establishing global sequence structure. When this global structure was violated, by either shortening the sequence or ending it with a locally expected but globally unexpected sound transition, activity slightly increased in VIP and PV neurons, respectively. Hence, distinct cellular mechanisms predict different regularity levels in sound sequences.

## Introduction

The ability to predict incoming sensory inputs at different timescales is a key cognitive function both in humans and animals (*Dehaene et al., 2015*; *Friston, 2005*; *Keller and Mrsic-Flogel, 2018*). Predictions are classically revealed by specific responses when a sensory event violates an expectation which stems either from the observation of regularities in passively observed stimulus sequences (*Bekinschtein et al., 2009*; *Carbajal and Malmierca, 2018*; *Gavornik and Bear, 2014*; *Grimm et al., 2016*; *Lao-Rodríguez et al., 2023*; *Näätänen et al., 1978*; *Squires et al., 1975*; *Ulanovsky et al., 2004*) or from the repeated experience of sensory outcomes produced by motor actions (*Attinger et al., 2017*; *Audette et al., 2022*; *Keller et al., 2012*; *Schneider et al., 2018*; *Solyga and Keller, 2024*). While the first evidence of sequence violation responses were based on the repetition of the same stimulus, e.g., in the so-called oddball or stimulus-specific adaptation protocols (*Farley et al., 2010*; *Lao-Rodríguez*

*et al., 2023*; *Näätänen et al., 1978*; *Squires et al., 1975*; *Taaseh et al., 2011*; *Ulanovsky et al., 2003*), a large range of experiments also revealed predictive processes for sequences of two or more stimuli (*Asokan et al., 2021*; *Bekinschtein et al., 2009*; *Gavornik and Bear, 2014*; *Keller et al., 2012*; *Tang et al., 2023*). Different levels of prediction complexity were conceptualized in parallel with these experiments, allowing to uncover the brain signatures of different predictive processing levels (*Wacongne et al., 2011*; *Parras et al., 2017*; *Hamm and Yuste, 2016*). The simplest form of prediction relates to the occurrence probability of a single stimulus, irrespective of other stimuli. Occurrence probability can be captured by stimulus-specific adaptation mechanisms, in which a variable, such as synaptic strength or neural excitability is updated by each stimulus occurrence and keeps a short-term memory of the occurrence thanks to a long relaxation time constant (*Ulanovsky et al., 2004*; *Tsodyks and Markram, 1997*; *Varela et al., 1997*). Numerous studies indicate that stimulus-specific adaptation mechanisms play an important role in the violation responses observed in oddball paradigms (*Chen et al., 2015*; *Hamm and Yuste, 2016*; *Parras et al., 2017*). At a higher complexity level, some predictions require the knowledge of the probability of a stimulus conditioned on the occurrence of one or more earlier events (*Balsam and Gallistel, 2009*). In many cases, these relationships can be effectively captured by local transition probabilities between successive stimuli (probability of stimulus N given the identity of stimulus N-1). In such cases, violation responses in the brain largely reflect violations of learned first-order local transition probabilities, as observed both in humans and non-human primates (*Kaposvari et al., 2018*; *Maheu et al., 2019*; *Meyer et al., 2014*; *Meyniel et al., 2016*). Finally, at an even higher level, some sequence structures are defined globally, meaning that the occurrence of a stimulus is fully predicted if and only if multiple previous stimuli are taken into account, possibly including information about the number of stimuli within identified chunks of stimuli interleaved by sequence interruptions (*Dehaene et al., 2015*). Hence, the ability to chunk sequences (*Ericcson et al., 1980*; *Fonollosa et al., 2015*; *Graybiel, 1998*) is an important part of predictive processing.

These different levels of regularity are all present in the local-global paradigm (*Bekinschtein et al., 2009*). In this paradigm, two distinct tones (denoted X and Y) are presented within short sequences of five tones (either XXXXX or XXXXY), which are themselves repeated multiple times in two different types of blocks: in one block (denoted XX for simplicity), the frequent sequence is XXXXX, and in the other block (denoted XY), the frequent sequence is XXXXY. This paradigm leads to both local and global expectations. The local expectation arises from the frequent transition X>X, which is violated at the end of XXXXY sequences. The global expectation arises from the frequent sequence, which is violated by occasionally presenting a rare unexpected sequence (XXXXX instead of XXXXY, or vice-versa). In XX block, hearing XXXXX XXXXX … XXXXY violates both a local and a global expectation, while in XY blocks, hearing XXXXY XXXXY … XXXXX violates only the global expectation that the sequence should terminate with a different sound Y. This paradigm also modulates stimulus probability (one sound X is more frequent than the other sound Y) and thereby produces stimulus-specific adaptation.

The local-global paradigm has been applied extensively to show that violations at different complexity levels produce brain responses with different cortical localization, timing, and sensitivity to changes in consciousness and attention. Global violations produce late responses in temporal and prefrontal cortices of awake humans and non-human primates, accompanied by top-down inputs to the auditory cortex (*Bekinschtein et al., 2009*; *Bellet et al., 2024*; *Chao et al., 2018*; *El Karoui et al., 2015*; *Mazancieux et al., 2023*; *Uhrig et al., 2016*; *Wacongne et al., 2011*). Those global violation responses are strongly affected when the subject is anesthetized (*Nourski et al., 2018*; *Tasserie et al., 2022*), sleeps (*Strauss et al., 2015*), or is instructed to divert his attention away from the stimuli (*Bekinschtein et al., 2009*). By contrast, stimulus-specific adaptation and responses to violations of local predictions (transition probabilities between successive stimuli) are fast, dominate in sensory cortices, largely resist inattention, sleep, or anesthesia, and can even occur in deep subcortical structures (*Parras et al., 2017*; *Tang et al., 2023*). This set of results suggests the existence of distinct mechanisms for a hierarchy of prediction levels. Yet, in the absence of a neural circuit-scale analysis of the hierarchy of violation responses, the neuronal mechanisms underlying these key predictive mechanisms remain largely elusive.

Here, we performed two-photon calcium imaging and Neuropixels electrophysiology in the auditory cortex of awake and anesthetized mice, during the local-global paradigm. We show that even

if responses purely attributable to global violations are weak, the auditory cortex captures different types of sequence violations based on several parallel mechanisms. Local transitions are signaled by an increase of the sound response amplitude magnified if the transition is globally unexpected. This modulation relies on two anesthesia-resistant mechanisms: stimulus-specific adaptation and an adaptation-independent surprise response operating on long-time scales. In parallel, VIP interneurons signal stimulus omissions and sequence terminations in an anesthesia-sensitive manner, while PV interneurons weakly signal global violation. Together, these results indicate that the mouse brain learns several levels of regularities in auditory sequences thanks to a set of mechanisms distributed in the local cortical circuits.

## Results
### Sequence violations are broadly signaled in the mouse auditory cortex

We adapted the local-global paradigm to awake, passively listening mice, which were head-fixed and held in a tube to allow for simultaneous two-photon calcium imaging of the auditory cortex (*Figure 1A*). With two distant pure tones, A (4 kHz) and B (12 kHz), we generated four different five-tone sequences starting with four repetitions of the same tone and terminating with a local sound transition (AAAAB, BBBBA) or with no transition (AAAAA, BBBBB). Tones within a sequence were 50 ms long, and their onsets were regularly spaced by a 237.5 ms time interval. We presented four different blocks of 125 sequences interspaced by 1.5 s of silence. In each block, the frequent sequence was first presented 25 times to create a global expectation, then, it was randomly replaced in 25 of its next 100 repetitions with an oddball sequence that differed only in the last stimulus (*Figure 1A*). In 10 of the 25 oddball sequences, the last sound was omitted and, in the 15 remaining oddball sequences, B was replaced by A or conversely.

Each mouse was presented with all four possible blocks (frequent sequence=AAAAA, AAAAB, BBBBB, or BBBBA), each repeated twice in a randomized fashion. Two blocks with common AAAA and BBBB sequences were used to control for rare omission sequences in the five-tone blocks. Two-photon calcium imaging of layer 2/3 neurons in the auditory cortex was performed continuously during each block yielding fluorescence variations produced by sounds across several hundred GCAMP6s expressing neurons across large 1 × 1 mm field-of-views (*Figure 1B and C*). In order to estimate firing rate modulations in the neurons, we used a simple linear deconvolution technique which is robust for indicators with a long decay time constant like GCAMP6s (*Deneux et al., 2019*; *Deneux et al., 2016*). This method yielded sufficient temporal resolution to partially isolate individual tone responses in a given five-tone sequence (*Figure 1C*). With this method, we collected responses to the four different sequences in each of the two possible conditions (common or rare) and to omission sequences for 8514 neurons across 12 sessions and four mice. Three different depths (150 μm, 250 μm, and 350 μm) were recorded in each mouse on three separate days.

Averaging deconvolved signals across all sampled neurons, we observed that the population response after the last sound was larger when a sequence is rare than when a sequence is common, but only for AAAAB and BBBBA sequences, which have a local transition at the last sound (*Figure 1D and E*). In the absence of a local transition (AAAAA or BBBBB, *Figure 1D and E*) or for omission sequence (*Figure 1—figure supplement 1*), the population response level was not significantly different for rare *vs* common sequences. Therefore, we observed a global violation response at the population level in the layer 2/3 neurons only when the global violation was coupled to a local violation (local + global) and not in absence of a local violation (pure global). The local + global violation response was seen in 10 to 20% of the recorded neurons (*Figure 1F*), which provided enough information to discriminate between rare and common sequences in single trials with cross-validated linear classifiers (*Figure 1G*). Intrigued by the absence of a response to the pure global violation seen in humans and non-human primates, all using electrophysiological techniques (*Bekinschtein et al., 2009*; *Chao et al., 2018*; *Uhrig et al., 2016*; *Wacongne et al., 2011*), we performed recordings using Neuropixels probes (*Figure 1H*), also targeted to mainly layer 2–3 (*Figure 1—figure supplement 2*). Theoretically, calcium imaging with a sensor expressed based on the synapsin promotor and extracellular recordings samples the same broad population of neurons. However, sampling biases across techniques are likely. For example, rapidly firing interneurons produce weak calcium signals that may not be efficiently detected in calcium imaging, while their dense firing is likely efficiently detected in

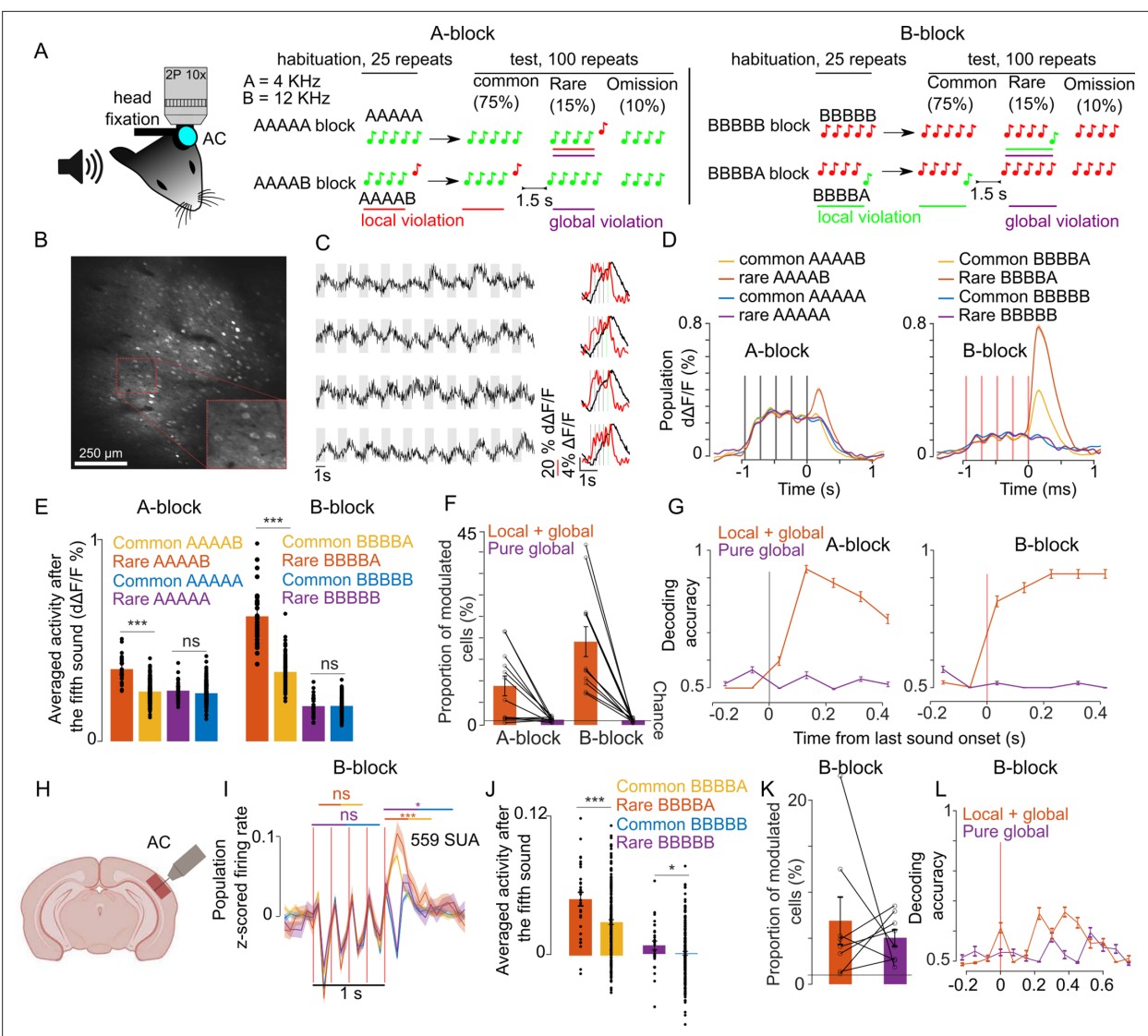

**Figure 1.** Sequence violations are broadly signaled in the mouse auditory cortex. (**A**) Sketch of the experimental setup and of the four different blocks of sound sequences. (**B**) Representative 1 × 1 mm two-photon imaging field-of-view with a magnification of the red rectangle. (**C**) Sample raw ΔF/F calcium traces (left, gray shadings=5 tone sequences) and trial-averaged activity during sequences (right black, raw ΔF/F; right red, temporally deconvolved ΔF/F=dΔF/F) for 4 neurons. (**D**) (left) Mean ± SEM population responses (dΔF/F) to rare or frequent AAAAB and AAAAA sequences. (right) Same for BBBBA and BBBBB sequences. (**E**) (left) dΔF/F population responses to the fifth tone averaged from 0 to 300 ms after tone offset for the AAAAX (left) and BBBBX (right) blocks. Mean + SEM & statistical tests: AAAAB rare vs common 0.36 ± 0.028% vs 0.25 ± 0.010% dΔF/F, p=5 × 10–14, AAAAB rare vs common 0.25 ± 0.022% vs 0.24 ± 0.008% dΔF/F, p=0.09; BBBBA rare vs common 0.62 ± 0.063% vs 0.34 ± 0.023% dΔF/F, p=8 × 10$^{-18}$; BBBBB rare vs common 0.17 ± 0.023% vs 0.18 ± 0.008% dΔF/F, p=0.54; Mann-Whitney U test; n=30 and n=180 for common and rare repetitions. (**F**) Proportion of cells that respond significantly more to rare than common AAAAB (9 ± 2.33%), AAAAA (1.24 ± 0.15%), BBBBA (19.4 ± 3.49%), or BBBBB (1.09% ± 0.14%, mean ± SEM, n=12 sessions, significance in each cell assessed with Mann-Whitney U test p<0.01, n=30 and n=180 for common and rare repetitions). (**G**) Performance of a cross-validated classifier, trained on time-averaged responses of 90 ms bins, for predicting rare vs common sequences. p<0.01 for all points above 50% (shuffle test 100 repetitions). (**H**) Sketch of electrophysiological recordings. (**I**) Mean ± SEM of z-scored single unit activities averaged across all units. (**J**) Mean single unit activity in the time window indicated by lines in (**I**) (rare vs common BBBBA 0.04 ± 0.006 vs 0.03 ± 0.002, p=8 × 10$^{-4}$; rare vs common BBBBB, 0.007 ± 0.004 vs 0.000 ± 0.001; p=0.03; Mann-Whitney U test; n=30 for rare and n=200 for common repetitions). (**K**) Same as (**F**) for electrophysiological recordings (BBBBA, 6.90 ± 2.57%, and BBBBB, 5.05 ± 0.90%, Mann-Whitney U test for each cell, n=8 sessions). (**L**) Same as (**G**) for electrophysiological recordings. p<0.05 for all points above 50% (shuffle test 100 repetitions).

The online version of this article includes the following figure supplement(s) for figure 1:

**Figure supplement 1.** Responses of synapsin-based expression of GCAMP6s to the omitted sequences in the auditory cortex.

**Figure supplement 2.** Probe localizations from Neuropixel recordings of the auditory cortex.

spike sorting. We, therefore, did not expect identical results from each methodology. We isolated 559 single units in eight recordings in four mice during which only BBBBA and BBBBB blocks were presented (see Materials and methods). Pooling together the activity of these units, we observed a clear local + global violation response consistent with calcium imaging (*Figure 1I–J*). However, surprisingly, we also observed a weak but significant pure global violation response appearing both at the population level (*Figure 1I–J*) and in a significant fraction of neurons (*Figure 1K*). This response occurred more than 200 ms after the last sound (*Figure 1I*), and provided enough information to discriminate between rare and common sequences above the chance level (*Figure 1k*).

## Local violations selectively modulate the responses of neurons tuned to the violating sound

To identify functional cell types involved in the signaling of sequence structure violations in a hypothesis-free manner, we performed a clustering analysis using the correlation between temporal response profiles across all conditions as a metric of functional similarity between neurons (see Materials and methods). Clustering was performed in a cross-validated manner, taking only a fraction of available trials to build the clusters and using the remaining fraction of trials to verify that the clustered response profiles were not emerging from the aggregation of similar noise patterns (*Figure 2A and B*) as occurred occasionally (e.g. *Figure 2B*). The clustering of calcium imaging data revealed diverse response types, with a small fraction of clusters showing suppression in response to A or B sounds and a larger fraction of clusters activated by A or B with more or less specificity (*Figure 2C*). No specific response to pure global violations (rare XXXXX) was observed. By contrast, most neuron clusters activated by A or B displayed an increased response to the preferred sound when this sound generated a local + global violation (rare XXXXY) as compared to when the violation was only local (common XXXXY) (*Figure 2C*), consistent with the population-level results (*Figure 1*). One exception to this rule was observed in cluster #30 (*Figure 2C*) which responds to the B tone only at the end of AAAAB sequences and not for BBBBX sequences. Clusters of isolated single units with boosted responses upon local + global violations were also observed in Neuropixels data (*Figure 2D*). In addition, Neuropixels data uncovered a large fraction of clusters suppressed during the sequence. This was in line with the dominance of suppressive response at the population level in this dataset (*Figure 1I*) and contrasted with calcium imaging data, indicating possible sampling biases across the two datasets. Notably, in two Neuropixel clusters, the suppressed response was followed by weak activating responses to pure global violations with a time course similar to the long-lasting response profiles of global violations responses observed in human MEG or EEG recordings (*Strauss et al., 2015*; *Wacongne et al., 2011*; *Figure 2E*). However, if some of these responses were statistically significant when individually tested, the significance level was above threshold when accounting for multiple testing across all extracted clusters. This further corroborates the observation that responses to global violations are weak in the mouse auditory cortex if not combined with a local violation.

## Modulation of sound responses by unexpected local violations is only partially due to adaptation

The large modulation of sound responses when a global violation is accompanied by a local violation (rare XXXXY) could be due to a stronger adaptation to the less frequent oddball sound (Y) in the control blocks where XXXXY sequences are repeated, compared to blocks where XXXXX sequences are repeated. To test this hypothesis, we reasoned that recovery from adaptation in the cortex is typically complete after 2–3 s (*Ulanovsky et al., 2004*). We, therefore, increased the inter-sequence interval to 28–30 s (*Figure 3A*) and repeated two-photon calcium imaging and Neuropixels recordings in awake mice. Specifically, we sampled 9441 neurons in 27 recordings in eight mice with two-photon imaging, and 559 single units in 8 recordings in four mice with Neuropixel (same dataset as for the 1.5 s interval). We observed that the larger responses at the end of XXXXY sequences are maintained despite the wide intersequence interval in calcium imaging (*Figure 3B–D*, *Table 1*) and Neuropixels (*Figure 3E*) data. This reflected a positive response modulation in neurons tuned to the oddball sound as revealed by a clustering analysis of single neuron responses (*Figure 3—figure supplements 1 and 2*). In both datasets, no specific response was observed for pure global violations (rare XXXXX, *Figure 3B–E*). The local + global effect for the ~30 s inter-sequence interval, was slightly weaker than for the 1.5 s inter-sequence interval in calcium imaging population responses (30 s: rare XXXXY=151.50

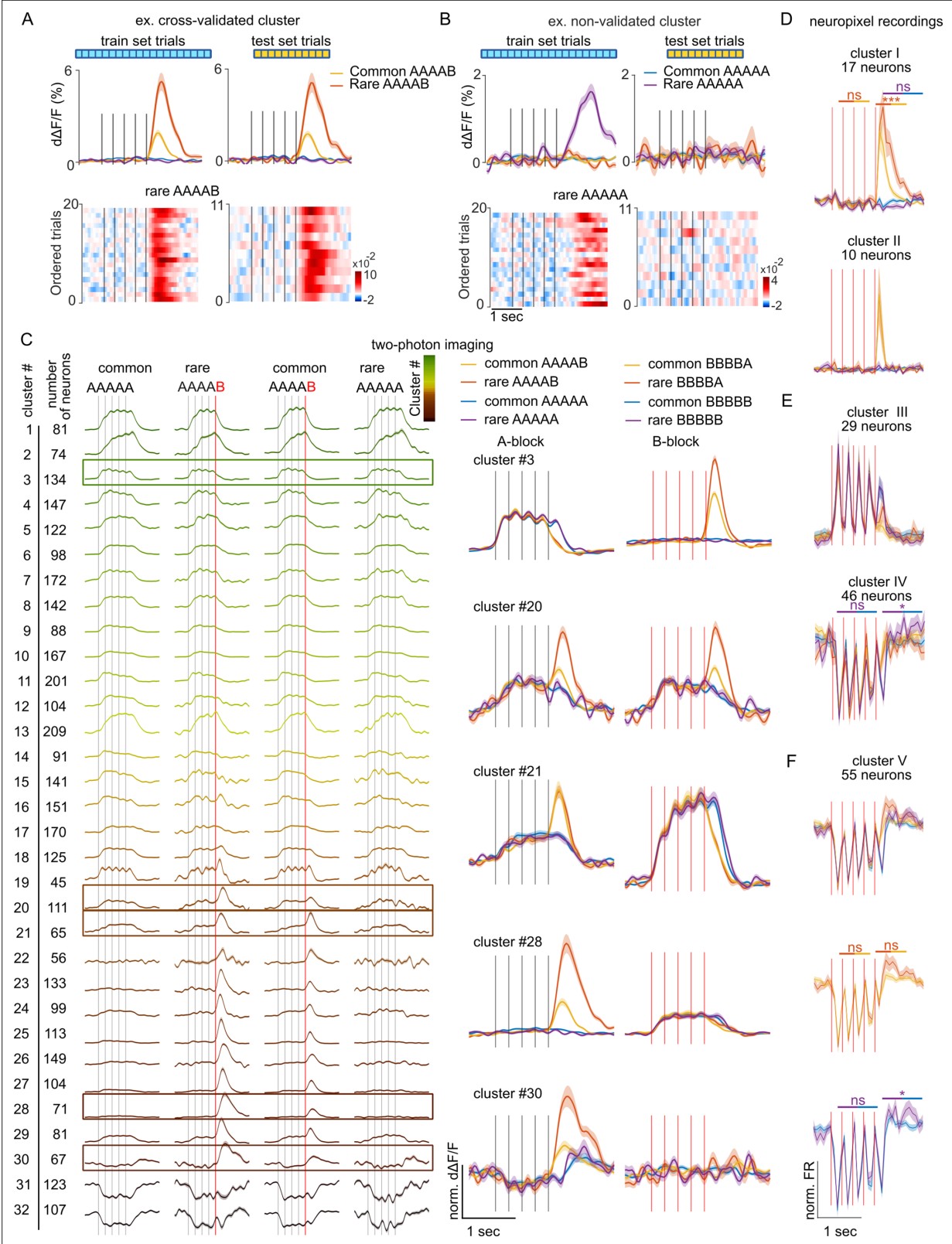

**Figure 2.** Local violations selectively modulate the responses of neurons tuned to the violating sound. (**A**) Sample cross-validated cluster with its dΔF/F response profile for the training (left) and test (right) sets (shading=SEM). Single-trial responses in test and training sets are shown below. (**B**) Same as (**A**) for a cluster that did not pass cross-validation. (**C**) AAAAX block responses of all cross-validated neuron clusters for the imaging dataset of *Figure 1*. The frame indicates clusters whose responses to AAAAX and BBBBX blocks are shown on the right. (**D**) Sample neuronal clusters for the

*Figure 2 continued on next page*

*Figure 2 continued*

electrophysiological recordings showing a local + global effect (cluster I and cluster II). (**E.F**) Responses of sample clusters with typical sound responses to both A and B tones and with a significant pure global effect (IV and V). All tests are Mann-Whitney U tests on the z-scored test set responses; n=10 rare and n=66 common sequence responses. Cluster I: offset (p=8 × 10⁻⁴) and onset (p=0.777) of rare vs common BBBBA, offset of rare vs common BBBBB (p=0.656). cluster IV: offset (p=0.038) and onset (p=0.913) of rare vs common BBBBB. cluster V: offset (p=0.383) and onset (p=0.907) of rare vs common BBBBA, offset (p=0.012) and onset (p=0.388) of rare vs common BBBBB.

---

± 7.95% common XXXXY; 1.5 s: rare XXXXY=182.15 ± 11.89% common XXXXY; p=0.0124, Wilcoxon rank sum test, n=12 sessions for 1.5 s and n=27 sessions for 30 s interval) compare *Figures 1E and 3C*. Accordingly, a smaller number of neurons were modulated by the global violation for the long inter-sequence interval even when correcting for the smaller number of rare sequences for the 30 s interval protocol (30 s: 8.66 ± 1.48%; 1.5 s: 16.55 ± 2.36%; p=3 × 10–3, Wilcoxon rank sum test, n=12 sessions for 1.5 s and n=27 sessions for 30 s interval, *Figure 3D*). Overall, those results suggest that only part of the local + global response for the 1.5 s inter-sequence interval is due to stimulus-specific adaptation.

To estimate the time scale on which stimulus specific adaptation plays a role in sound response modulations, we performed two-photon calcium imaging of 593 auditory cortex neurons (two recordings in one mouse) during a classical oddball sound paradigm in which a standard tone is repeated with a fixed inter-tone interval, and occasionally replaced with deviant tone (*Figure 3F*). Varying systematically the inter-tone interval, we observed larger oddball responses compared to when the same tone is a standard up to a 2 s interval, but not for 4 s or larger intervals (*Figure 3F*). This indicates that recovery from adaptation is complete after 4 s. To corroborate this, we performed two-photon calcium imaging while presenting in the same imaging sessions two blocks with BBBBA or BBBBB sequences repeated at a 30 s interval, occasionally violated by BBBBB and BBBBA sequences, respectively, and two blocks within which the A tone was presented with the same time intervals as in the BBBBA and BBBBB blocks but in the absence of B tones (*Figure 3G*). We observed in 6906 neurons collected in eight recordings in three mice that the modulation of responses to rare A's occurred only in the presence of the B tones associated with A in the repeated sequences (*Figure 3G*). Together, these results show that responses to local + global violations correspond to a surprise mechanism that cannot be entirely explained by adaptation. This, therefore, suggests that the local + global violation response reflects a prediction about the short time scale structure of the 1 s XXXXX sequence chunks rather than about the occurrence frequency of the oddball sound Y.

## Unexpected local violations produce a surprise response based on a recently experienced sequence structure

To further disentangle the contribution of adaptation and of the sequence-based predictions in violation responses, we reasoned that adaptation corresponds to a dampening of the response to a specific stimulus over frequent repeats, whereas the violation of a prediction should rather correspond to an increase of the response for stimuli that are unlikely to occur based on the predictable context. We, therefore, plotted neural responses for each sequence repetition across the full protocols with 1.5 s and ~30 s time intervals (*Figure 4A and B*). For the 1.5 s interval, we observed a reproducible diminution of the response across the first three XXXXY sequences when common (*Figure 4A and C*) which reflects adaptation. For the 30 s interval, we observed no decrease in the responses to the Y tone across successive repeats of the XXXXY sequence (*Figure 4B and D*). Instead, responses to Y when not predictable based on sequence context were always larger than when Y is regularly repeated at the end of the sequence (*Figure 4B*), corroborating the idea that they reflect a surprise response to an unexpected transition rather than adaptation. For the 1.5 s interval, we also observed a significant diminution of the end response to the rare XXXXY sequences across a single block (*Figure 4A and C*). These observations indicate the coexistence of three types of phenomena: (i) stimulus-specific adaptation, that we quantified as the difference between the first and last response to a stimulus in a block, (ii) surprise, which we quantified as the difference between the responses to the first rare and first common XXXXY, and therefore reflects the elevation of XXXXY response when XXXXX is predicted, (iii) surprise adaptation, which is the diminution of the surprise response upon frequent surprises (first - last rare XXXXY). As shown in *Figure 4E*, these three phenomena coexist when the sequences are repeated at a short interval, but only the surprise component is maintained for the long inter-sequence interval. This surprise response, similar to adaptation (*Nieto-Diego and Malmierca, 2016*;

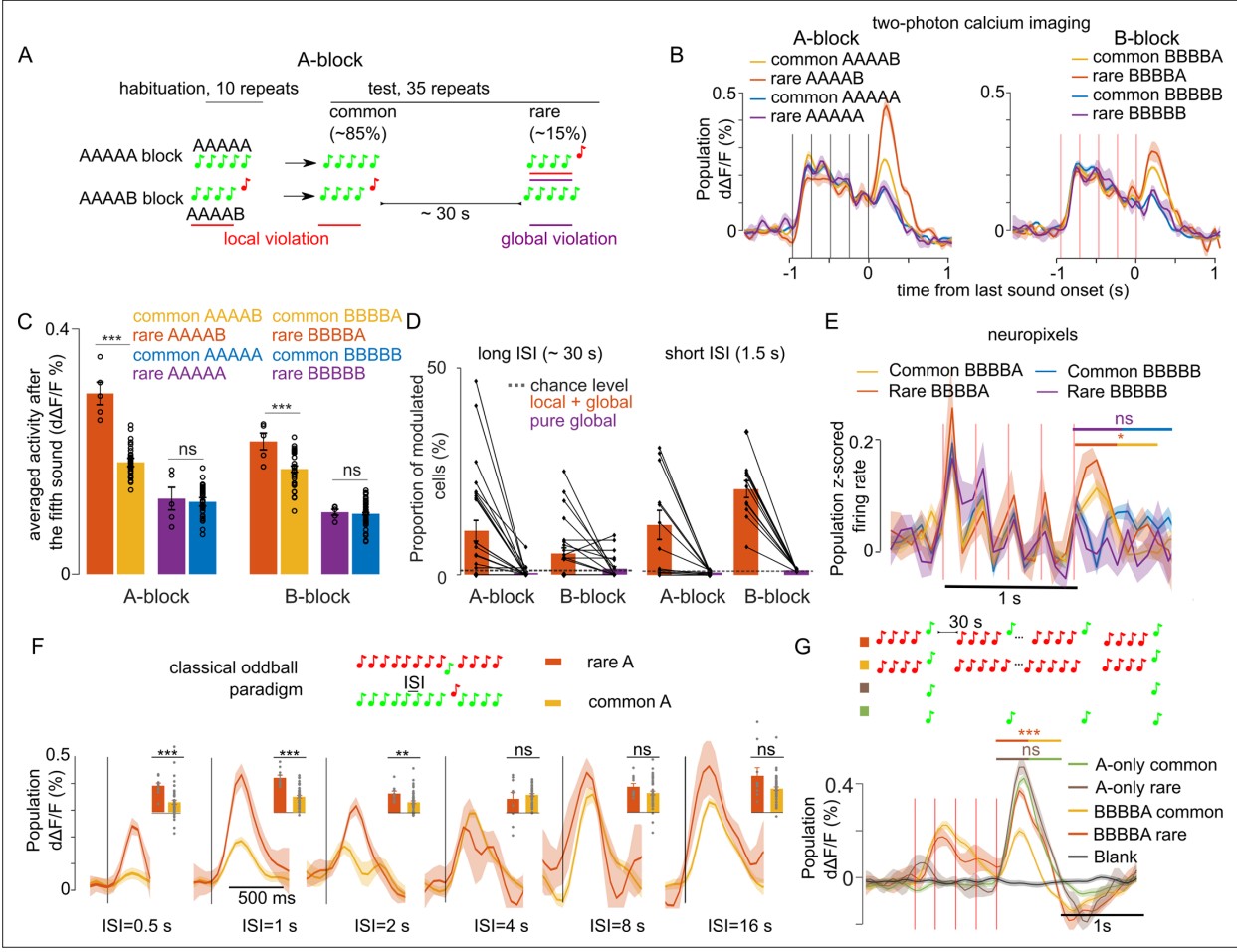

**Figure 3.** Modulation of sound responses by unexpected local violations is only partially due to adaptation. (**A**) Sketch of the experimental paradigm. (**B**) (left) Trial-averaged population responses (dΔF/F) to rare and common AAAAB or AAAAA sequences (right). Same for the BBBBA and BBBBB sequences. (**C**) Time-averaged (0–300 ms after 5th tone offset) population responses to the fifth tone, rare vs common. AAAAB: 0.29 ± 0.018% vs 0.18 ± 0.006% dΔF/F, p=2 × 10−4; AAAAA: 0.12 ± 0.018 vs 0.12 ± 0.007% dΔF/F, p=0.41; BBBBA: 0.22 ± 0.013% vs 0.17 ± 0.005% dΔF/F, p=6 × 10−3; BBBBB: 0.10 ± 0.005% vs 0.09 ± 0.002% dΔF/F, p=0.40; Mann-Whitney U test; n=5 for rare and n=40 for common sequences. (**D**) Proportion of sound responsive cells that are significantly (Mann-Whitney U-test, p<0.01) upward modulated at the end of rare sequences (see Materials and methods) for long (~30 s, right) and short (1.5, left) inter-sequence intervals (ISI). The number of sequence repeats was equalized across datasets (n=5 for rare and n=40 for common sequence). Long ISI (27 sessions), AAAAB: 10.61 ± 2.51% and AAAAA: 0.40 ± 0.26% BBBBA: 5.10 ± 1.40% and BBBBB: 1.44 ± 0.54%. Short ISI (12 sessions), AAAAB: 12.12 ± 3.51%, AAAAA: 0.60 ± 0.12%; BBBBB: 20.78 ± 2.04%, BBBBB: 1.16 ± 0.08%. (**E**) Same as (**B**) for z-scored electrophysiological recordings (for statistics see *Figure 3—figure supplement 2*). (**F**) Mean ± SEM population responses to rare (orange) and common (yellow) in classical stimulus-specific adaptation (SSA) paradigm using single tones instead of sound sequences. (*p<0.05, **p<0.01, ***p<0.001; Mann-Whitney U test; see *Table 1* for detailed statistics.) (**G**) Mean ± SEM population responses to common BBBBA (yellow) and rare BBBBA (orange) in BBBBX blocks and to A tones played at the same time intervals as in the BBBBX blocks but excluding all B tones. Green: frequent. Brown: rare. Gray: blank sound probe. Statistics are computed on the time-averaged response, 0–500 ms after A tone onset: common BBBBA 0.08 ± 0.01% dΔF/F, rare BBBBA: 0.20 ± 0.01% dΔF/F, p=0.10, Mann-Whitney U test; n=6 for rare and n=43 for common sequences. A-only (frequent) 0.23 ± 0.01% dΔF/F, (rare) 0.27 ± 0.02; p=2 × 10−4; Mann-Whitney U test; n=5 for rare and n=42 for frequent repetitions.

The online version of this article includes the following figure supplement(s) for figure 3:

**Figure supplement 1.** Clustering of neuronal responses for the long inter-sequence interval, dataset 1.

**Figure supplement 2.** The same neurons exhibit local + global violation responses for both short and long intersequence intervals in wakefulness but not under anesthesia (although it is present at the population level).

**Figure supplement 3.** The local + global effect is preserved across wakefulness and anesthesia.

**Table 1.** Mean ± SEM population responses to rare (orange) and common (yellow) in classical stimulus-specific adaptation (SSA) paradigm in *Figure 3*.

p-values are from the statistical tests comparing between rare and common conditions (Mann-Whitney U test; n=10 for rare and n=47 for common repetitions).

Stimulus specific adaptation paradigm

| Inter-stimulus intervals (second) | Mean (+SEM) dAF/F rare (%) | Mean (+SEM) dAF/F common (%) | p-value |
|---|---|---|---|
| 0.5 | 0.193 (=0.019) | 0.073 (+0.015) | $1 \times 10^{-4}$ |
| 1 | 0.252 (+0.019) | 0.114 (+0.010) | $1 \times 10^{-4}$ |
| 2 | 0.140 (+0.017) | 0.076 (+0.010) | 0.002 |
| 4 | 0.100 (+0.048) | 0.130 (+0.009) | 0.545 |
| 8 | 0.185 (+0.023) | 0.139 (+0.014) | 0.055 |
| 16 | 0.266 (+0.058) | 0.172 (+0.013) | 0.073 |

*Taaseh et al., 2011*), is maintained during anesthesia (*Figure 3—figure supplement 3*), suggesting that it is a pre-attentive phenomenon.

## VIP interneurons signal sequence termination and omission of expected local violations

We next investigated how vasoactive intestinal peptide-positive (VIP) interneurons in the auditory cortex represent sequence violations. VIP neurons were shown to signal stimulus local omissions of image repeats in the visual cortex (*Garrett et al., 2020*), and are, therefore, important candidates for violation signals in the auditory cortex. We specifically expressed GCAMP6s by local injections of a floxed AAV-GCAMP6s virus in the auditory cortex of Vip-IRES-cre mice (*Figure 5A*). We imaged 3547 VIP neurons across 15 sessions, using a local-global protocol that included oddball sequences with replacement or omission of the last sound. The inter-sequence interval was 1.5 s. Strikingly, the population of VIP neurons was globally suppressed by all sequences with a late rebound activity after the sequence whose amplitude was significantly larger for two particular violations: for an unexpected local transition in rare AAAAB sequences (*Figure 5B*) and for the omission of B in blocks where AAAAB is the standard sequence (*Figure 5C*). In blocks where B was the standard tone, local violation responses were weak and we could not detect responses to omission, probably due to the fact that we sampled more extensively VIP neurons preferring tone A. Violation signals provided enough information to decode above chance level if a sequence was rare or common in blocks where A was the standard tone (*Figure 5E*). Interestingly, the strong response suppression disappeared under isoflurane anesthesia (*Figure 5D*), as well as violation responses for omissions (AAAA in AAAAB block: –0.0010 ± 0.002% dΔF/F, AAAA in AAAA control block: 0.0006% dΔF/F±0.001%; p=0.707; Mann-Whitney U test; n=20 for rare and n=100 for common repetitions), suggesting a strong re-organisation of VIP neuron activity during anesthesia. The population response profile, however, concealed a diversity of neuronal responses across VIP neurons which we could uncover by clustering neurons according to their response time courses (*Figure 5F*). While many VIP neurons were suppressed (*Figure 5F*, cluster 4), others were driven by sounds (*Figure 5F*, clusters 1, 2, 3), and some were silent (*Figure 5f*, cluster 7). Most remarkably, a small but salient population of VIP interneurons signaled the termination of sequences (*Figure 5F*, cluster 5). Termination responses were characterized by a slight suppression during all sequences followed by a large activity increase of about 200 ms after the last sound. When the sequence was shortened by one sound the termination response occurred earlier, showing that it follows the actual sequence structure and not a global expectation based on repeated sequence length. Moreover, the termination signal was similar across all sequences although slightly larger for unexpected omissions (*Figure 5F*, cluster 5), consistent with population observations (*Figure 5C*). During anesthesia many sound-driven, suppressed, and sequence termination neurons became unresponsive (*Figure 5F*). By contrast, some VIP neurons that were silent in the awake state became responsive under anesthesia, confirming the massive re-organisation of VIP neuron activity in this state (*Figure 5D and F*).

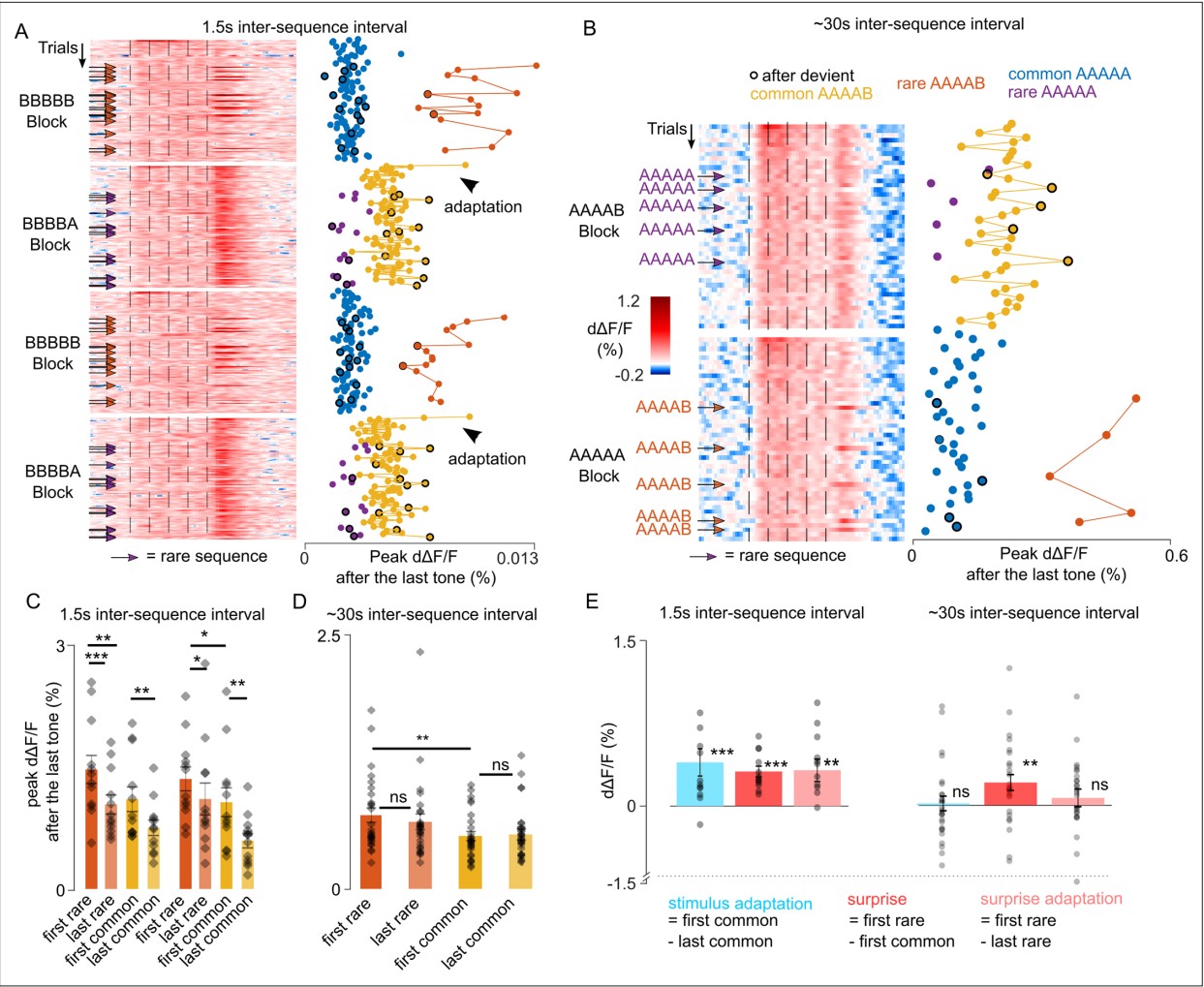

**Figure 4.** Unexpected local violations produce a surprise response based on recently experienced sequence structure. (**A**) (left) Heatmap of population responses of ordered single trials for two different blocks with short inter-sequence intervals (ISI) (arrows indicate rare stimuli). (right) Mean population responses to the fifth tone of each trial (0–300 ms from tone onset). (**B**) Same as A, for the long ~30 s ISI. (**C**) Time-averaged population responses to the fifth tone for common and rare sequences, comparing responses of the first and to the last trial of each condition, and measured for the first (left) and second repetition of each block. Statistics 1st block repetition: first rare: 1.466 ± 0.173%, last rare: 1.038 ± 0.117%, p=4 × 10⁻⁴; first common: 1.102 ± 0.151%, last common: 0.751 ± 0.095%, p=0.06 (first rare vs first common, p=0.003); Statistics 2ns block repetition: first rare: 1.351 ± 0.146%, last rare: 1.105 ± 0.196%, p=0.038; first common: 1.065 ± 0.177%; last common: 0.592% ± 0.089%; p=0.008 (first rare vs first common, p=0.010). Wilcoxon signed-rank test; n=12 sessions. (**D**) Same as (**C**) for the ~30 s ISI (Mean ± SEM): first rare: 0.724% ± 0.070%, last rare: 0.658 ± 0.077%, p=0.055; first common: 0.519% ± 0.044%, last common: 0.534 ± 0.055%, p=0.328 (first rare vs first common, p=0.003). Wilcoxon signed-rank test; n=27. (**E**) Short ISI - stimulus adaptation effect: 0.411 ± 0.128%; p=0.003; surprise: 0.325 ± 0.050%; p=4 × 10⁻⁴; surprise adaptation: 0.337 ± 0.105%, p=0.009; Wilcoxon signed-rank test; n=12. Long ISI; stimulus adaptation effect = 0.015 ± 0.066 %, p=0.648; surprise: 0.204 ± 0.072%; p=0.005; surprise adaptation: 0.065 ± 0.08%; p=0.107. Wilcoxon signed-rank test; n=27.

## PV interneurons weakly signal pure global violations

We finally imaged parvalbumin positive (PV) interneurons during the local global paradigm with the 1.5 s inter-sequence interval (**Figure 6A**). Sampling 2215 neurons across seven sessions, we observed larger population responses at the end of AAAAB sequences when they were rare than when they were common (**Figure 6B**). This confirmed that local + global violation responses are largely broadcasted and also involve PV neurons. No response to omissions was observed in PV interneurons (**Figure 6C**). Interestingly, we observed a larger response at the end of AAAAA sequences when they were rare than when they were common, and this effect disappeared under anesthesia (**Figure 6D**). Although weak, the information provided by global violation responses in the awake state sufficed to discriminate rare from common AAAAA sequences at the single trial level based on a population

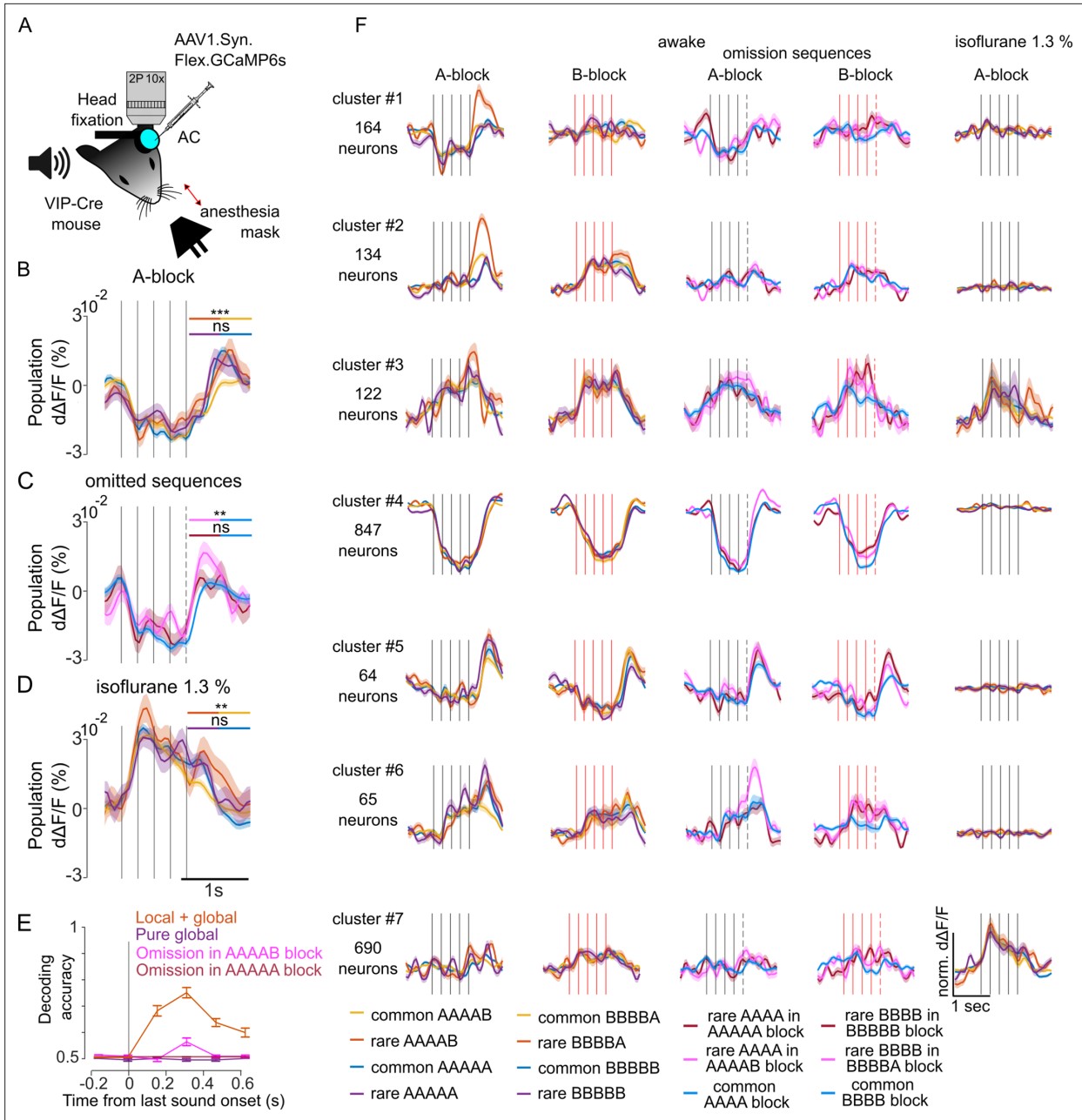

**Figure 5.** Vasoactive intestinal peptide positive (VIP) interneurons signal sequence termination and omission of expected local violations. (**A**) Sketch of the experimental setup. (**B**) Mean ± SEM population responses to the rare and common AAAAB and AAAAA sequences. Statistics for the time-averaged fifth tone response: AAAAB rare: 0.003 ± 0.002% dΔF/F; AAAAB common: –0.003 ± 0.0007% dΔF/F; p=5 × 10⁻⁴; AAAAA rare 0.002 ± 0.002% dΔF/F, AAAAA common: 0.0008 ± 0.0008% dΔF/F, p=0.29, Mann-Whitney U test, n=30 for rare and n=180 for common repetitions. (**C**) Mean ± SEM population responses to AAAA sequence in the AAAAA (omission, red), AAAAB (omission, pink) or AAAA block (control, blue). Statistics comparing omission to control conditions: 0.003 ± 0.003% dΔF/F, 0.006 ± 0.002% dΔF/F, –0.0001 dΔF/F ± 0.0009%; omission of A in AAAAA, p=0.292; omission of B in AAAAB, p=0.008; Mann-Whitney U test; n=20 for rare and n=100 for common repetitions. (**D**) Same as (**B**) under isoflurane anesthesia. AAAAB rare: 0.010 ± 0.002% dΔF/F, AAAAB common 0.003 ± 0.0008% dΔF/F; p=0.0056, AAAAA rare 0.006 ± 0.001% dΔF/F, AAAAA common: 0.004 ± 0.0008% dΔF/F, p=0.123, Mann-Whitney U test; n=30 for rare and n=180 for common repetitions. (**E**) Performance of a fully cross-validated classifier for predicting the rare sequence against the common sequence. The classifier is trained on time-averaged responses of 160 ms bins. p<0.01 for all points above 50% (shuffle test 100 repetitions). (**F**) Cross-validated clusters of neurons. Mean ± SEM responses during wakefulness and under anesthesia.

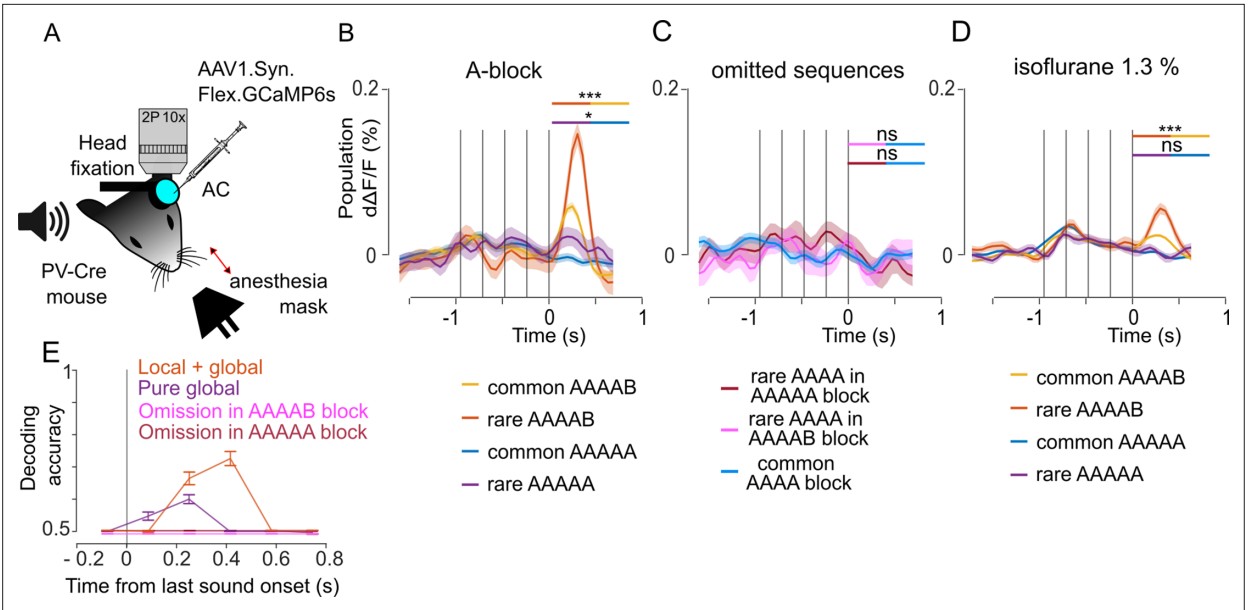

**Figure 6.** Parvalbumin positive (PV) interneurons weakly signal pure global violations. (**A**) Sketch of the experimental setup. (**B**) Mean ± SEM population responses (deconvolved calcium signals) to rare and common AAAAB and AAAAA sequences, AAAAB rare: 0.058 ± 0.009% dΔF/F, AAAAB common: 0.021 ± 0.003% dΔF/F, p=1 × 10$^{-4}$; AAAAA rare: 0.012 ± 0.009%; AAAAA common: –0.007 ± 0.003% dΔF/F, p=0.025, Mann-Whitney U test; n=30 for rare and n=180 for common sequences. (**C**) Mean ± SEM population responses to AAAA sequence in the AAAAA (omission, red), AAAAB (omission, pink) or AAAA block (control, blue). Statistics comparing omission to control conditions: –0.009 ± 0.011% dΔF/F, –0.008 ± 0.009% dΔF/F; –0.007 ± 0.004% dΔF/F, omission of A in AAAAA, p=0.6176; omission of B in AAAAB, p=0.523; Mann-Whitney U test; n=20 for rare and n=100 for common repetitions. (**D**) Same as (**B**) under isoflurane anesthesia/ AAAAB rare: 0.030 ± 0.003% dΔF/F, AAAAB common, 0.015 ± 0.001% dΔF/F, p=1× 10$^{-4}$; AAAAA rare: 0.002 ± 0.003% dΔF/F, AAAAA common 0.002 ± 0.001% dΔF/F, p=0.535, Mann-Whitney U test, n=30 for rare and n=180 for common sequences. (**E**) Performance of a fully cross-validated classifier for predicting the rare sequence against the common sequence. The classifier is trained on time-averaged responses of 160 ms bins. p<0.01 for all points above 50% (shuffle test, 100 repetitions).

classifier (*Figure 6E*). Together, these results indicate that PV interneurons contribute to the signaling of global violations, even when they are not accompanied by a local violation.

## Discussion

In this study, we characterized auditory cortex activity during the local-global paradigm to investigate the mechanisms underlying the predictive processing of recurring auditory sequences. Our results indicate first that the most salient effect of sequence structure is to modulate sound-specific responses depending on the predictability of perceived sounds. The same sound triggers larger responses when it is weakly predictable than when it is highly predictable (*Figure 1*). In line with observations made over several decades with the classical oddball paradigm (*Nelken, 2014*), part of this modulation is due to stimulus-specific adaptation, which corresponds to a progressive decrease in responsiveness upon repetition of a stimulus. This decrease was clearly exhibited by deviant sounds in the local-global protocol when the intersequence interval was 1.5 s (*Figure 4*). Adaptation level is incremented by stimulus repeats occurring within its typical recovery time. Therefore, it is a suitable process to estimate the occurrence probability of a sound through a running average over a short time scale, independent of the presence of other sounds (at least) in the absence of cross-stimulus adaptation which occurs when sounds are similar (*Yarden et al., 2022*). The occurrence probability of a sound can also be deduced from the local transition probability between sounds (*Meyniel et al., 2016*). The existence of processes performing such predictions in the mouse brain is demonstrated by our observation that a repeated local transition between an X and a Y sound leads to a lower response to Y than when the transition is unexpected, even over a timescale that does not allow for an occurrence probability estimate via adaptation (*Figure 3*). This adaptation-independent and anesthesia-resistant violation response aligns with previous observations of stronger responses to deviant stimuli in mice trained with repeated visual stimulus sequences (*Gavornik and Bear, 2014*; *Tang et al., 2023*) although in

these cases the protocol does not allow disentangling the violation of transition probabilities from stimulus-specific and cross-stimulus adaptation. Our results, therefore, clearly indicate that at least two parallel processes estimate stimulus occurrence probability in the sensory cortex: one estimating the probability of a sound independently of other sounds and one making an estimation of conditional probability relative to the previous sound (at least). The modulations of sound response resulting from the violation of these predictions are cumulative and involve the same sound-responsive neurons (*Figure 4*; *Figure 3—figure supplements 1 and 2*).

In addition to these two processes which modulate sensory responses to signal local changes in the sequence, we have also observed several types of anesthesia-sensitive responses that specifically signal more global aspects of the expected sequence structure or of its violations. The first and most robust response type was observed in a subgroup of VIP-positive interneurons which elevate their firing rate about 200 ms after the end of a sequence, irrespective of the sequence type (*Figure 5*). These responses differ from offset responses in the auditory cortex by their long delay and their lack of specificity. These neurons were a small fraction of all VIP neurons, about 2% (*Figure 5*), and therefore are a rare cell type. They signal the interruption of systematic sound repeats and could serve to mark the border of sequence chunks. These responses could be considered to signal the omission of the next sound in a regular sequence of sounds irrespective of the specific sequence content. Their response pattern is similar to the chunking signals that are observed in the basal ganglia and prefrontal cortex and mark the beginning and end of action sequences (*Barnes et al., 2005*; *Fujii and Graybiel, 2003*).

In parallel with these termination responses, we observed responses of the population of VIP interneurons to omissions of the B sound in repeated AAAAB sequences, but not to omissions of A in repeated AAAAA sequences and not to omissions in BBBBA blocks (*Figure 5*). At the single neuron level, these omission responses reflected a small rebound of neurons that were suppressed during the sequence, or more rarely, a boosting of a ramping response during the sequence (*Figure 5*). The absence of omissions responses when XXXXX is the expected sequence suggests that the omission detection mechanism requires a salient expectation. As we did not observe a salient response to A in BBBBA sequences in the sampled population of VIP neurons, we speculate that the absence of omission responses for the absence of A in BBBBA sequence was due to weak signaling of A in the considered VIP population. Omission responses have been recently observed through multielectrode recordings in the auditory cortex of mice (*Lao-Rodríguez et al., 2023*). Our results suggest that the omission-responsive neurons in this study may be VIP neurons. The sensitivity of omission responses to anesthesia reported in this study is also in line with our observations (*Figure 5*). Moreover, populations of VIP interneurons have been implicated in the signaling of omitted images in sequences of regularly repeated identical images by ramping activity until the next image (*Garrett et al., 2020*), suggesting that VIP interneurons share the function of signaling omissions of salient predictions across sensory cortex.

The last type of sequence violation feature observed in our experiment is the anesthesia-sensitive slight boosting of PV-positive interneurons at the end of AAAAA sequences when AAAAB sequences are expected, i.e., a 'pure global' effect which was also seen in Neuropixel recordings (*Figures 1 and 6*). Although this effect provides enough information to discriminate between rare and common AAAAA sequences above chance level, it is much weaker than the modulation of sound response for unexpected XXXXY sequences. Moreover, we did not observe it for rare BBBBB sequences in our dataset. The weakness of the response suggests that the mismatch between the local continuation of a sound repeat and the global expectation of a sound transition makes this type of violation difficult to detect for the mouse brain, at least within the auditory cortex. The primary reason may be that global regularities are encoded at a higher level of the cortical hierarchy, with only weak top-down signals returning back to sensory cortices. In prior human and non-human primate experiments, robust responses to pure global violations were observed in, the prefrontal cortex, but also in the auditory cortex in mesoscopic-scale recordings (*Bekinschtein et al., 2009*; *Chao et al., 2018*; *Wacongne et al., 2011*) with late timing and functional connectivity suggesting that the latter arose from top-down signals (*Chao et al., 2018*). Our imaging approach covers a broad area of the auditory cortex (repeatedly sampled with a 1 × 1 mm field of view), and yet this was insufficient to observe a salient response to pure global violations. This may be because the networks necessary to capture the global regularity level are underdeveloped in mice compared to humans. Alternatively, it is possible that

these violation signals originate from a diffuse source in the auditory cortex (rare neurons or incoming axons) which was not sampled in our approach. Larger-scale experiments in mice, encompassing the prefrontal/cingulate cortex, will be necessary to discriminate between these hypotheses.

Overall, our results demonstrate a remarkable diversity of neural processes tracking the regularity of sound sequences in the mouse auditory cortex, some of them pre-attentive and robust to anesthesia, and others which are anesthesia-sensitive. Our results are in line with the general view that predictive processes are constantly ongoing in brain circuits, as suggested by the predictive coding theory (*Dayan et al., 1995*; *Friston, 2005*; *Keller and Mrsic-Flogel, 2018*; *Rao and Ballard, 1999*; *Srinivasan et al., 1982*). However, they also suggest that there is more than a single canonical circuit which performs predictions and detects violations. For example, our results highlight that the boosting of responses to unexpected sounds is both due to an item-specific adaptation and to a surprise response that takes into account transition probabilities and, for some neurons, the more global sequence structure. As they affect the same neurons, the coexistence of these two effects could only be detected by using an appropriate sequence structure and multiple time scales (*Figures 1–4*). Moreover, the classical predictive coding theory (*Friston, 2005*) postulates that omission and stimulus mismatch responses result from a single process in which top-down predictions are compared to bottom-up sensory signals. Our observation that omission, sequence termination responses, and pure global violation responses are produced by partially distinct interneurons and are anesthesia-sensitive while local transition mismatch responses are broadcasted across all cell types and resist anesthesia, indicates that mismatch and omissions are computed by different sub-circuits. Our results largely extend recent observations that different cell types have different dynamics of violation responses (*O'Toole et al., 2023*). Therefore, theories of how sound sequences generate predictions in the brain would need to incorporate an array of mechanisms addressing different aspects of the sequence structure rather than a single generic mechanism.

## Materials and methods

### Cranial window implantation and viral injections

All procedures were conducted in accordance with protocols approved by the French Ethical Committees #59 and #89 (authorizations APAFIS#9714–2018011108392486 v2 and APAFIS#27040–2020090316536717 v1). We used 8–12 wk-old C57BL/6 J (RRID:IMSR_JAX:000664), Vip-IRES-cre (*Vip*^tm1(cre)Zjh/J, RRID:IMSR_JAX:010908), and Pvalb-IRES-Cre (*Pvalb*^tm1(cre)Arbr/J, RRID:IMSR_JAX:017320) male and female mice housed 1–7 per cage, in normal light/dark cycle (12 hr/12 hr). Cranial window implantation and viral injections were performed under ketamine medetomidine or under isoflurane anesthesia (1.3–1.7%) with body temperature maintained constant at 37 °C using a thermal blanket. Part of the right masseter was surgically removed to expose the temporal bone. A craniotomy of 5 mm in diameter was drilled over the auditory-cortex on the right hemisphere. Three injections of 150 nl of AAV1.Syn.GCaMP6s or AAV1.Syn.Flex.GCaMP6s (~1 × $10^{-12}$ vg.ml$^{-1}$), obtained from Addgene and Vector Core (Philadelphia, PA, USA), were performed with glass micropipettes and a programmable oil-based injector (Nanoliter 2000 & Micro 4; World Precision Instruments) at 30 nl.min$^{-1}$. We targeted all subfields of the auditory cortex, including non-primary regions. The craniotomy was sealed with a glass window comprising a circular coverslip (5 mm diameter, pre-sealed with cyanoacrylate glue), and a metal post for head-fixation was implanted using two dental cements: Super-Bond C&B (Sun Medical Co. Ltd.) directly on the bone and Orthojet (Lang Dental, Wheeling, Illinois) for final sealing of the cranial window and the fixation post. Mice were given 1 wk to recover from the surgery. Imaging was performed between 4–7 wk after virus injection.

### Sounds and stimulation protocols

Sounds were generated using MATLAB and delivered at 192 kHz with a NI-PCI-6221 card (National Instruments), which fed an amplified free-field loudspeaker (SA1 and MF1-S, Tucker-Davis Technologies, Alachua, FL). The head-fixed mouse in a tube was isolated from external noise sources by a soundproof box that provided 30 dB attenuation across the 1–100 kHz range. The two-photon microscope used was a fully silent acousto-optic system.

We generated two pure tones at 4 and 12 kHz with an intensity of 70 dB SPL calibrated at the animal's ear position using a probe microphone (Brüel & Kjær, type 4939 L-002) over a duration of

50 ms including 10 ms linear intensity up- and down-ramp to avoid onset and offset artifacts. The 4 and 12 kHz tones were labeled as A and B, respectively, and combined in sequences of 4–5 tones with 237.5 ms time intervals in-between tone onsets. We used four sound stimulation protocols. A local-global protocol with short inter-sequence (2.5 s) intervals (*Figures 1–2 and 4–5–6*), a local-global protocol with long inter-sequence (28–30 s) intervals (*Figure 3*), a classical oddball protocol with several inter-stimuli intervals (*Figure 3F*) and a single-tone (A-only) repetition protocol followed by a local-global protocol with long (30 s) inter-sequence intervals (*Figure 3G*).

The short interval protocol included 10 blocks of ~5 min corresponding to 125 sequences of 1 s separated by a fixed 1.5 s silence period. Eight blocks consisted of one common five-tone sequence repeated 25 times alone and then 75 times randomly interleaved with a rare five-tone sequence, repeated 15 times, and with a four-tone omission sequence repeated 10 times. These eight blocks correspond to four different block types repeated each twice: block type 1 (common AAAAB, rare AAAAA, rare omission AAAA), block type 2 (common AAAAA, rare AAAAB, rare omission AAAA), block type 3 (common BBBBA, rare BBBBB, rare omission BBBB), block type 4 (common BBBBB, rare BBBBA, rare omission BBBB). One additional block type included 50 AAAA sequences followed by a 62.5 s silence period and then 50 BBBB sequences. This block type was repeated twice and was used as a comparison with oddball AAAA or BBBB omission sequences.

The long interval protocol included four blocks of 45 sequences each separated by a 28–30 s interval between two sequences. The 45 sequences consisted of one common 5-tone sequence repeated 10 times during the habituation phase and then 30 times randomly interleaved with rare sequences of 5-tone, repeated five times. There were four different block types repeated only once: block type 1 (common AAAAB, rare AAAAA), block type 2 (common AAAAA, rare AAAAB), block type 3 (common BBBBB, rare BBBBA), block type 4 (common BBBBA, rare BBBBB). In one experiment (*Figure 3B*), we delivered a white noise burst (70 dB SPL, 50 ms duration including 10 ms intensity ramps) with a random 3–5 s interval after each sound sequence followed by a silence period of 25 s. We observed no difference in the response observed during protocols including white noise bursts and those without.

In the electrophysiological recordings (*Figure 1H–I*, *Figure 3E*, *Figure 3—figure supplements 2 and 3*) and one of the two-photon calcium imaging experiments (*Figure 3—figure supplements 2 and 3*), we used a protocol containing both short and long inter-sequence intervals, but only for the two blocks of B-block with two types of sequences (BBBBB and BBBBA sequences). We recorded the same neurons using short (1.5 s) and long (30 s) interval protocols in awake animals, then only the long (30 s) interval protocol in anesthetized animals. It should be noted that during electrophysiological recording, we were unable to match the same single units between awake and anesthetized animals contrary to the two-photon calcium imaging.

The classical stimulus-specific adaptation (SSA) protocol included 12 blocks of 67 sounds each separated with six different inter-stimuli intervals of 0.5 s; 1 s; 2 s; 4 s; 8 s; and 16 s. For each interval, there were two different block types repeated only once. A block begins with a habituation phase of 10 standard tones (tone A) and is followed by a random repetition of 47 standard tones (tone A) and 10 deviant tones (tone B). In the second block of each interval, the deviant and standard tones were switched.

The single-tone protocol (A only) included two blocks and was followed by two additional blocks of local-global paradigm with a long interval between sequences (30 s). Blocks 3 and 4 contained the repetition of 50 sequences separated by a 30 s interval. At the beginning of the block, 10 standard sequences (BBBBA or BBBBB) were played, followed by a randomized repetition of 34 standard sequences and six rare sequences (BBBBB or BBBBA, respectively). During the first two blocks, we delivered only the last deviant tone of the sequences, and instead of the BBBBB sequence, which contains no deviant tone, blank stimuli were presented (*Figure 3G*). The order of the blocks was changed between rarely presented and commonly presented stimuli. We had two types of recordings: (1) A only rare - A only common - BBBBA rare - BBBBA common. (2) A only common - A only rare - BBBBA common - BBBBA rare that we combined. In all the data collected in this study, it was systematically observed that the initial tone of each session generated particularly high activity. To avoid this effect we excluded the first tone of each session here.

## Two-photon calcium imaging in wake and anesthetized mouse auditory cortex

One week before imaging, mice were trained to stand still, head-fixed under the microscope for five consecutive days for 15 min to 1 hr per day. Then mice were imaged for 1 hr long sessions with up to four vertical depths imaged per mouse on different days. Imaging was performed using a two-photon microscope (Femtonics, Budapest, Hungary) equipped with an 8 kHz resonant scanner combined with a pulsed laser (MaiTai-DS, SpectraPhysics, Santa Clara, CA, USA) tuned at 920 nm. The objective was a 10 x Olympus (XLPLN10XSVMP), obtaining a field of view of 1000 × 1000 µm. Images were acquired at 31.5 Hz during trials of 315.5 s. For the anesthesia, VIP, and PV interneuron experiments, two-photon imaging was performed with a fully silent acousto-optic microscope (Karthala) combined with a pulsed laser (Insight, Spectra Physics). The objective was a 16 x (N16XLWD-PF, Nikon). Images were acquired from four planes at 19.1 Hz per plane interleaved by 50 µm with fields of view of 478 × 478 µm. In the short interval protocol, calcium activity was acquired continuously during an entire block with the Femtonics microscope and during half a block with the Karthala microscope (the interruption between two half blocks was below 3 s). In the long interval protocol, calcium activity was recorded during the sequence and until 1 s after the white noise presentations (between –1 s to –2 s and 4.5 s to 7.5 s from sequence onset).

The anesthesia sessions followed the awake sessions, without modifying the field of view. During anesthesia sessions, a nose mask from the SomnoSuite anesthesia unit (Kent Scientific) was used to induce narcosis. An infrared heating pad (Kent Scientific) was placed under the mouse tube. In the beginning, the isoflurane level was gradually increased to 2.5% for 2 min to induce narcosis. The anesthesia was then gradually reduced to around 1.3% (range 1.1–1.4%) during recordings, adjusting based on observed whisker movements and pupil size.

## Calcium imaging data analysis

Motion artifacts, regions of interest selection, and the signal extraction were carried out using the Python-based version of Suite2p (*Pachitariu et al., 2017*). Then, data analysis was performed using custom Matlab scripts. Neuropil contamination was subtracted by applying the following equation: $F_{cor}(t)=F(t) – 0.7\ F_n(t)$. Then the change in fluorescence $\Delta F/F_0$ was computed as $(F_{cor}(t) - F_0)/F_0$, where $F_0$ is estimated as the minimum of Gaussian filtered calcium trace, for each block. $\Delta F/F_0$ was then temporally deconvolved to yield a more accurate estimate of neuronal firing rate changes, using a linear algorithm using the following formula: $r(t) = \Delta F/F_0'(t) + \Delta F/F_0(t) / \tau$ in which $\Delta F/F_0'$ is the first temporal derivative of $\Delta F/F_0$ and $\tau$ the calcium decay time constant which we set to 2 s for GCaMP6s. After deconvolution, a Gaussian smoothing filter ($\sigma$=1.5 or 2 frames) was applied to the data.

## Acute extracellular recordings in wake and anesthetized mouse auditory cortex

Electrophysiology was performed using Neuropixels 1.0 probes (384 channels) on four mice that had previously been used for calcium imaging through a glass coverslip placed above the auditory cortex. For track reconstruction, the electrodes were dipped in diI, diO or diD (Vybrant Multicolor Cell-Labeling Kit, Thermo Fisher) prior to recording and allowed to dry at least 15 min before insertion. Prior to recording, a small hole was drilled in the coverglass to allow probe access, and the electrode was slowly inserted into the brain and left in position for 20 min before beginning recording to stabilize and minimize movements of neurons. Recordings were performed using a warmed cortex buffer filling a small kwik cast well formed around the edge of the coverslip to retain the liquid and in contact with the reference electrode. After each recording, the surface of the brain was amply flushed and then protected with Kwik-Cast. Two recordings were performed per animal. Data was sampled at 30 kHz using a NI-PXI chassis (National Instruments) and the SpikeGLX acquisition software.

After five of the awake sessions, anesthesia sessions were conducted without moving the Neuropixel probes, using the same protocol as for anesthesia sessions for two-photon calcium imaging.

## Spike sorting and analysis of extracellular recordings

Electrophysiological signals were high-pass filtered and spike sorting was performed using the CortexLab suite (https://github.com/cortex-lab, UCL, London, England). Single unit clusters were identified using kilosort 2.5 (https://github.com/cortex-lab/KiloSort; *Pachitariu et al., 2016*) followed

by manual corrections based on the interspike-interval histogram and the inspection of the spike waveform using Phy (https://github.com/cortex-lab/phy; *Rossant, 2020*).

Single-trial sound responses were extracted (0.5 s before and 2 s after sound onset) with 75 ms time bins.

## Cross-validated clustering analysis

Single trial responses to each five or four-tone sequence were extracted from the raw deconvolved traces including a 0.5 s baseline and a 1 s post-sequence period for each neuron. We averaged a subset of trials (training set), pooling trials from the two repetitions of each block type. For each sequence, we considered whether it occurred commonly or rarely in different conditions. Clustering was performed by using the average response signatures of each cell to selected sets of conditions as described in the figure legends. We used agglomerative hierarchical clustering (*Deneux et al., 2019*; *Deneux et al., 2016*) based on the Ward method to group together neurons with similar response signatures. The similarity metric used was the Pearson correlation between response profiles. A threshold was applied to the resulting dendrogram to obtain N clusters based on the number of neurons and conditions in each dataset. The threshold is chosen so that increasing the number of clusters does not lead to an increase in the number of different response types in the clustering. These clusters were then manually sorted to remove all obvious groups of non-responsive cells as indicated by an absence of activity above or below the typical baseline noise level for the cluster. Finally, the responsive clusters were tested on the remaining trials, which were not used for the clustering (test set). If we observe the same response profile as in the train test, we have validated the cluster (e.g. *Figure 2A*). This cross-validation ensures that clusters do not simply aggregate noise patterns (e.g. *Figure 2B*).

## Cross-validated classifiers

In order to evaluate if violation responses provide sufficient information to differentiate between a rare and a common sequence, we used a decoder applied to all neurons of a given dataset pooled across mice and recording sessions, and cross-validated using a leave-one-out procedure. To reduce dimensionality and improve classifier training, we use clusters of neurons instead of individual neurons. Next, we train a linear SVM classifier to discriminate between the common and rare conditions. Finally, we test the classifier using the left-out trial for all averaged 90 ms time bins describing the response to the sequence, including the training time bin. Then another trial is left out and the procedure is performed again. This is repeated until all rare and common trials are tested. The performance of the classifier in each time bin is given as the average of the classifier output (0=wrong, 1=correct classification) for all test trials balanced by the number of trials per condition.

## Fraction of neurons selective to a condition

We evaluated the fraction of neurons in each session selective to the rare or common presentation of a sequence in the time span between 0–300 ms after the last tone onset. For each cell, we computed the p-value of a Wilcoxon rank sum test evaluating the probability that means responses in the considered time bin are greater for the rare than common condition (for the short-interval protocol, we remove the first 10 repetitions of the common stimuli from the habituation phase of each block). Using an alpha-value of 0.01, we then computed the fraction of cells with a p-value below the alpha value. This fraction represents the proportion of neurons showing a significant difference in response between rare and common conditions during the 300 ms time window after the last tone onset (*Figure 1F and K*).

To compare the short (*Figure 1D*) and long (*Figure 3B*) datasets, we first chose an SNR threshold common to both datasets above which cells are retained, in order to extract cells with a higher signal-to-noise ratio (SNR=mean activity divided by its standard deviation). This threshold (=0.2) was determined on the basis of the SNR calculated from the first four tones of the sequences, defining responsiveness to a specific frequency. This threshold allows us to contain between 5% and 12% of neurons in each data set. We then correct our statistical test so that the number of repetitions N of the rare and common stimuli in the short-interval protocol is identical to that in the long-interval protocol. To do this, we calculate the percentage of modulated neurons in the short-interval protocol by randomly selecting only 40 repetitions from 180 repetitions of the common condition and five

repetitions from 30 repetitions of the rare condition, repeat this process 100 times and calculate the mean percentage.

## Statistical analysis

All quantifications and statistical analyses were performed using Matlab scripts. We conducted statistical assessment using non-parametric tests reported in box plots, figure legends, and *Table 1*, including mean and standard error of the mean (SEM), number of samples, and nature of sample (number of sessions or trials). Significance levels were set at 5%.

## Code availability

The code used for the analysis are available on *Zenodo* https://doi.org/10.5281/zenodo.14027003.

## Acknowledgements

This work was supported by the European Research Council (ERC CoG 770841) and Fondation pour l'Audition (FPA IDA02, RD-2023–1, APA 2016–03), the Fondation pour la Recherche Médicale (SPF202005011970), and the Agence National pour la Recherche (ANR Infernoise). We acknowledge the support of the Fondation pour l'Audition to the Institut de l'Audition.

## Additional information

### Competing interests

Brice Bathellier: Reviewing editor, eLife. The other authors declare that no competing interests exist.

### Funding

| Funder | Grant reference number | Author |
| --- | --- | --- |
| European Research Council | ERC CoG 770841 | Brice Bathellier |
| Fondation Pour l'Audition | FPA IDA02 | Brice Bathellier |
| Fondation pour la Recherche Médicale | SPF202005011970 | Sophie Bagur |
| Agence Nationale de la Recherche | Infernoise | Sophie Bagur Brice Bathellier |
| Fondation Pour l'Audition | RD-2023-1 | Brice Bathellier |
| Fondation Pour l'Audition | APA 2016-03 | Brice Bathellier |

The funders had no role in study design, data collection and interpretation, or the decision to submit the work for publication.

### Author contributions

Sara Jamali, Conceptualization, Data curation, Software, Formal analysis, Investigation, Visualization, Methodology, Writing – review and editing; Sophie Bagur, Investigation, Methodology, Writing – review and editing; Enora Bremont, Investigation; Timo Van Kerkoerle, Conceptualization, Supervision, Project administration, Writing – review and editing; Stanislas Dehaene, Conceptualization, Supervision, Methodology, Project administration, Writing – review and editing; Brice Bathellier, Conceptualization, Resources, Data curation, Formal analysis, Supervision, Funding acquisition, Validation, Investigation, Methodology, Writing – original draft, Project administration, Writing – review and editing

### Author ORCIDs

Sara Jamali http://orcid.org/0000-0003-4933-3167
Sophie Bagur https://orcid.org/0000-0002-4375-1838
Timo Van Kerkoerle https://orcid.org/0000-0003-1935-8216
Brice Bathellier https://orcid.org/0000-0001-9211-1960

### Ethics

All procedures were conducted in accordance with protocols approved by the French Ethical Committees #59 and #89 (authorizations APAFIS#9714-2018011108392486 v2 and APAF-IS#27040-2020090316536717 v1).

Reviewer #1 (Public review): https://doi.org/10.7554/eLife.102702.2.sa1
Reviewer #2 (Public review): https://doi.org/10.7554/eLife.102702.2.sa2
Reviewer #3 (Public review): https://doi.org/10.7554/eLife.102702.2.sa3
Author response https://doi.org/10.7554/eLife.102702.2.sa4

### Data availability

All data and analysis code used in this study are available on Zenodo without access restriction, https://doi.org/10.5281/zenodo.14027003.

The following dataset was generated:

| Author(s) | Year | Dataset title | Dataset URL | Database and Identifier |
|---|---|---|---|---|
| Jamali S, Bagur S, Brémont E, Van Kerkoerle T, Dehaene S, Bathellier B | 2024 | Data from: Parallel mechanisms signal a hierarchy of sequence structure violations in the auditory cortex | https://doi.org/10.5281/zenodo.14027003 | Zenodo, 10.5281/zenodo.14027003 |

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
