## [Editor Report · eLife Assessment]

This **important** study advances our understanding of the way neurons in the auditory cortex of mice respond to unpredictable sounds. Through the use of state-of-the-art recording methods, **compelling** evidence is provided that responses to local and global violations in sound sequences are prediction errors and not simply the consequence of stimulus-specific adaptation. Although the cell-type-specific results are intriguing, further work is needed to establish their reliability.

---

## [Referee Report · Reviewer #1 (Public review)]

Summary:

The authors successfully detected distinct mechanisms signalling prediction violations in the auditory cortex of mice. For this purpose, an auditory pure-tone local-global paradigm was presented to awake and anaesthetised mice. In awake rodents, the authors also evaluated interneuron cell types involved in responses to the interruption of the regularity imposed by local-global sequences. By performing two-photon calcium imaging and single-unit electrophysiology, the authors disentangled three phenomena underlying responses to violations of the distinct local-global regularity levels: Stimulus-specific adaptation, surprise and surprise adaptation. Both stimulus-specific adaptation and surprise-or deviant-evoked responses are observable

under anaesthesia. Altogether, this work advances our understanding of distinct predictive processes computing prediction violations upon the complexity of the regularity imposed by the auditory sequence.

Strengths:

it is an elegant study beautifully executed.

Weaknesses:

No weaknesses were identified by this reviewer.

---

## [Referee Report · Reviewer #2 (Public review)]

Summary:

Oddball responses are increases in sensory responses when a stimulus is encountered in an unexpected location in a sequence of predictable stimuli. There are two computational interpretations for these responses: stimulus-specific adaptation and prediction errors. In recent years, evidence has accumulated that a significant part of these sequence violation responses cannot be explained simply by stimulus-specific adaptation. The current work elegantly adds to this evidence by using a sequence paradigm based on two levels of sequence violations: "Local" sequence violations of repetitions of identical stimuli, and "global" sequence violations of stimulus sequence patterns. The authors demonstrate that both local and global sequence violation responses are found in L2/3 neurons of the mouse auditory cortex. Using sequences with different inter-stimulus intervals, they further demonstrate that these sequence violation responses cannot be explained by stimulus-specific adaption.

Strengths:

The work is based on a very clever use of a sequence violation paradigm (local-global paradigm) and provides convincing evidence for the interpretation that there are at least two types of sequence violation responses and that these cannot be explained by stimulus-specific adaption. Most of the conclusions are based on a large dataset, and are compelling.

Weaknesses:

The final part of the paper focuses on the responses of VIP and PV-positive interneurons. The responses of VIP interneurons appear somewhat variable and difficult to interpret (e.g. VIP neurons exhibit omission responses in the A block, but not the B block). The conclusions based on these data appear less solid.

---

## [Referee Report · Reviewer #3 (Public review)]

Summary:

In their manuscript entitled "Parallel mechanisms signal a hierarchy of sequence structure violations in the auditory cortex", Jamali et al. provide evidence for cellular-level mechanisms in the auditory cortex of mice for the encoding of predictive information on different temporal and contextual scales. The study design separates more clearly than previous studies between the effects of local and global deviants and separates their respective effects on the neuronal responses clearly through the use of various contextual conditions and short and long time scales. Further, it identifies a contribution from a small set of VIP interneurons to the detection of omitted sounds, and shows the influence of isofluorane anesthesia on the neural responses.

Strengths:

(1) The study provides a rather encompassing set of experimental techniques to study the cellular level responses, using two complementary recording techniques in the same animal and similar cortical location.

(2) Comparison between awake and anesthetized states are conducted in the same animals, which allows for rather a direct comparison of populations under different conditions, thus reducing sampling variability.

(3) The set of paradigms is well developed and specifically chosen to provide appropriate and meaningful controls/comparisons, which were missing from previous studies.

(4) The addition of cell-type specific recordings is valuable and in particular in combination with the contrast of awake and anesthetized animals provides novel insights into the cellular level representation of deviant signals, such as surprise, prediction error, and general adaptation.

(5) The analysis and presentation of the data are clear and quite complete, yet remain succinct and perform insightful contrasts.

(6) The study will have an impact on multiple levels, as it introduces important variations in the paradigm and analytical contrasts that both human and animal researchers can pick up and improve their studies. The cell-type-specific results are particularly intriguing, although these would likely require replication before being completely reliable. Further, the study provides a substantial and diverse dataset that others can explore.

Weaknesses:

(1) The responses from cells recorded via Neuropixel and 2p differ qualitatively, as noted by the authors, with NP-recorded cells showing much more inhibited/reduced responses between acoustic stimulations. The authors briefly qualify these differences as potentially indicating a sampling issue, however, this matter deserves more detailed consideration in my opinion. Specifically, the authors could try to compare the different depths at which these neurons were sampled or relate the locations in the cortex to each other (as the Neuropixel recordings were collected in the same animals, a subset of the 2p recordings could be compared to the Neuropixel recordings.).

(2) The current study did not monitor the attentional state of the mouse in relation to the stimulus by either including a behavioral component or pupil monitoring, which could influence the neural responses to deviant stimuli and omissions. .

(3) Given the complexity and variety of the paradigms, conditions, and analyzed cell-types, the manuscript could profit from a more visual summary figure that provides an easy-to-access overview of what was found.

---

## [Author Response]

Author response:

**Reviewer #1 (Public review):**
Summary:The authors successfully detected distinct mechanisms signalling prediction violations in the auditory cortex of mice. For this purpose, an auditory pure-tone local-global paradigm was presented to awake and anaesthetised mice. In awake rodents, the authors also evaluated interneuron cell types involved in responses to the interruption of the regularity imposed by local-global sequences. By performing two-photon calcium imaging and single-unit electrophysiology, the authors disentangled three phenomena underlying responses to violations of the distinct local-global regularity levels: Stimulus-specific adaptation, surprise and surprise adaptation. Both stimulus-specific adaptation and surprise-or deviant-evoked responses are observable under anaesthesia. Altogether, this work advances our understanding of distinct predictive processes computing prediction violations upon the complexity of the regularity imposed by the auditory sequence.Strengths:it is an elegant study beautifully executed.Weaknesses:No weaknesses were identified by this reviewer.
**Reviewer #2 (Public review):**
Summary:Oddball responses are increases in sensory responses when a stimulus is encountered in an unexpected location in a sequence of predictable stimuli. There are two computational interpretations for these responses: stimulus-specific adaptation and prediction errors. In recent years, evidence has accumulated that a significant part of these sequence violation responses cannot be explained simply by stimulus-specific adaptation. The current work elegantly adds to this evidence by using a sequence paradigm based on two levels of sequence violations: "Local" sequence violations of repetitions of identical stimuli, and "global" sequence violations of stimulus sequence patterns. The authors demonstrate that both local and global sequence violation responses are found in L2/3 neurons of the mouse auditory cortex. Using sequences with different inter-stimulus intervals, they further demonstrate that these sequence violation responses cannot be explained by stimulus-specific adaption.Strengths:The work is based on a very clever use of a sequence violation paradigm (local-global paradigm) and provides convincing evidence for the interpretation that there are at least two types of sequence violation responses and that these cannot be explained by stimulus-specific adaption. Most of the conclusions are based on a large dataset, and are compelling.Weaknesses:The final part of the paper focuses on the responses of VIP and PV-positive interneurons. The responses of VIP interneurons appear somewhat variable and difficult to interpret (e.g. VIP neurons exhibit omission responses in the A block, but not the B block). The conclusions based on these data appear less solid.

We agree with the referee that the response modulations observed in VIP and PV-Positive interneurons are weak and variable. This is indicated in the manuscript. Probably, larger scale recordings are necessary to ascertain fully the presence and distribution of omission responses.

**Reviewer #3 (Public review):**
Summary:In their manuscript entitled "Parallel mechanisms signal a hierarchy of sequence structure violations in the auditory cortex", Jamali et al. provide evidence for cellular-level mechanisms in the auditory cortex of mice for the encoding of predictive information on different temporal and contextual scales. The study design separates more clearly than previous studies between the effects of local and global deviants and separates their respective effects on the neuronal responses clearly through the use of various contextual conditions and short and long time scales. Further, it identifies a contribution from a small set of VIP interneurons to the detection of omitted sounds, and shows the influence of isofluorane anesthesia on the neural responses.Strengths:(1) The study provides a rather encompassing set of experimental techniques to study the cellular level responses, using two complementary recording techniques in the same animal and similar cortical location.(2) Comparison between awake and anesthetized states are conducted in the same animals, which allows for rather a direct comparison of populations under different conditions, thus reducing sampling variability.(3) The set of paradigms is well developed and specifically chosen to provide appropriate and meaningful controls/comparisons, which were missing from previous studies.(4) The addition of cell-type specific recordings is valuable and in particular in combination with the contrast of awake and anesthetized animals provides novel insights into the cellular level representation of deviant signals, such as surprise, prediction error, and general adaptation.(5) The analysis and presentation of the data are clear and quite complete, yet remain succinct and perform insightful contrasts.(6) The study will have an impact on multiple levels, as it introduces important variations in the paradigm and analytical contrasts that both human and animal researchers can pick up and improve their studies. The cell-type-specific results are particularly intriguing, although these would likely require replication before being completely reliable. Further, the study provides a substantial and diverse dataset that others can explore.Weaknesses:(1) The responses from cells recorded via Neuropixel and 2p differ qualitatively, as noted by the authors, with NP-recorded cells showing much more inhibited/reduced responses between acoustic stimulations. The authors briefly qualify these differences as potentially indicating a sampling issue, however, this matter deserves more detailed consideration in my opinion. Specifically, the authors could try to compare the different depths at which these neurons were sampled or relate the locations in the cortex to each other (as the Neuropixel recordings were collected in the same animals, a subset of the 2p recordings could be compared to the Neuropixel recordings.).

We agree with the referee that the sampling issue, which we propose as a possible explanation for the large difference between our Neuropixel and 2P imaging recordings, must be investigated more thoroughly. This is, however, largely outside of the scope of this study. We have reported the depth and location of Neuropixel recordings in Figure S2. The depth is similar for both techniques covering mostly layers 2, 3 and 4. The location spans mostly the primary auditory cortex for two photon imaging and Neuropixel recordings. One Neuropixel recording is located in the ventral secondary auditory cortex. We could not find any evidence that the response to global violations in Neuropixel data stems specifically from this particular recording.

(2) The current study did not monitor the attentional state of the mouse in relation to the stimulus by either including a behavioral component or pupil monitoring, which could influence the neural responses to deviant stimuli and omissions.

As reported by Bekinschtein et al. 2009, the attentional state influences responses to global violation in human subjects. It is extremely difficult to precisely compare attentional states in mice and human subjects. We have performed recordings in mice that had to attend to sound to detect a white noise sound in between the sequence to obtain a reward. This did not lead to increased global violation response. However, as the sequence themselves did not predict reward in this context they may divert attention. Therefore, this result is inconclusive and not worth including in our manuscript. If the sequence predicts rewards, there is a potential confound between violation responses and reward expectations or motor preparation signals. Pupil monitoring could be an alternative which we did not investigate.

(3) Given the complexity and variety of the paradigms, conditions, and analyzed cell-types, the manuscript could profit from a more visual summary figure that provides an easy-to-access overview of what was found.

This is an excellent suggestion, although given the complexity and diversity of our observations it may be hard to fit everything in one understandable figure.